# Rapid Glacier Shrinkage and Glacial Lake Expansion of a China-Nepal Transboundary Catchment in the Central Himalayas, between 1964 and 2020

Yan Zhong [1,2,3] , Qiao Liu [1,3,*] , Liladhar Sapkota [1,2], Yunyi Luo [1,2], Han Wang [1,2], Haijun Liao [1,2] and Yanhong Wu [1,3]

1 Institute of Mountain Hazards and Environment, Chinese Academy of Sciences, Chengdu 610041, China; zhongyan19@mails.ucas.ac.cn (Y.Z.); liladhar.767517@cdg.tu.edu.np (L.S.); luoyunyi20@mails.ucas.ac.cn (Y.L.); wanghan202@mails.ucas.ac.cn (H.W.); liaohaijun@imde.ac.cn (H.L.); yhwu@imde.ac.cn (Y.W.)
2 College of Resources and Environment, University of Chinese Academy of Sciences, Beijing 100049, China
3 Branch of Sustainable Mountain Development, Kathmandu Center for Research and Education, Chinese Academy of Sciences-Tribhuvan University, Chengdu 610041, China
\* Correspondence: liuqiao@imde.ac.cn; Tel.: +86-135-4788-7225

**Abstract:** Climate warming and concomitant glacier recession in the High Mountain Asia (HMA) have led to widespread development and expansion of glacial lakes, which reserved the freshwater resource, but also may increase risks of glacial lake outburst floods (GLOFs) or debris floods. Using 46 moderate- and high-resolution satellite images, including declassified Keyhole and Landsat missions between 1964 and 2020, we provide a comprehensive area mapping of glaciers and glacial lakes in the Tama Koshi (Rongxer) basin, a highly glacierized China-Nepal transnational catchment in the central Himalayas with high potential risks of glacier-related hazards. Results show that the $329.2 \pm 1.9$ km$^2$ total area of 271 glaciers in the region has decreased by $26.2 \pm 3.2$ km$^2$ in the past 56 years. During 2000–2016, remarkable ice mass loss caused the mean glacier surface elevation to decrease with a rate of $-0.63$ m a$^{-1}$, and the mean glacier surface velocity slowed by ~25% between 1999 and 2015. The total area of glacial lakes increased by $9.2 \pm 0.4$ km$^2$ (~180%) from $5.1 \pm 0.1$ km$^2$ in 1964 to $14.4 \pm 0.3$ km$^2$ in 2020, while ice-contacted proglacial lakes have a much higher expansion rate (~204%). Large-scale glacial lakes are developed preferentially and experienced rapid expansion on the east side of the basin, suggesting that in addition to climate warming, the glacial geomorphological characters (aspect and slope) are also key controlling factors of the lake growing process. We hypothesize that lake expansion will continue in some cases until critical local topography (i.e., steepening icefall) is reached, but the lake number may not necessarily increase. Further monitoring should be focused on eight rapidly expanding proglacial lakes due to their high potential risks of failure and relatively high lake volumes.

**Keywords:** climate change; glacier change; glacial lake change; GLOFs; central Himalaya; Tama Koshi

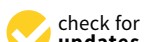



## 1. Introduction

Glaciers play essential roles in the global cryosphere and Earth's water cycle [1] by buffering freshwater supply [2], modifying mountain landforms [3], and regulating fluvial sediment flux [4]. Glacier and glacier-related processes are highly vulnerable to climate change [5,6]. The Tibet Plateau and its surroundings, namely the High Mountain Asia (HMA), host the largest volume of glaciers outside of the polar regions [7,8] and is referred to as the Asian water tower [9,10]. Among all, approximately 20,431 (covering 19,679 km$^2$) glaciers and 4950 (covering 455.3 km$^2$) glacial lakes are developed along with the Himalayas [11–14]. Nearly 800 million people in the Indus, Ganges and Yarlung Tsangpo river (i.e., Brahmaputra River) basins are benefited from the Himalayan glacial meltwaters [15,16].

With a warming global climate, Himalayan glaciers have shrunk severely [16–18], accelerating the development and evolution of glacial lakes in this region [19]. The existence of glacial lakes could cause a lag effect of glacial runoff that affects downstream river discharges, hydrological processes, channel erosions, and landscape evolutions [20]. They could also induce glacier-related hazards, e.g., glacial lake outburst floods [19], which is one of the main climate hazards caused by warming in high altitude regions [21]. Himalayan glacial lakes are mainly located between 4000 and 5700 m a.s.l., and expanded by approximately 14.1% from 1990 to 2015 [11]. The widespread expansion of the Himalayan glacial lakes has been the focus of attention by both scientific researchers and social communities [19,22–30].

Owing to frequent geo-hazards, regional development delays, poor road conditions, and low access to glaciers and glacial lakes, field investigations are difficult to conduct in the HMA. However, with the recent rapid development of remote sensing technology and cloud-based geospatial data distribution, efficient and reliable mapping methods have been widely used in cryosphere science [31,32]. To better assess and mitigate the risks of GLOFs hazard and their impact, it is crucial to investigate the evolution of the glaciers and glacial lakes and predict their future development trends under climate change [33]. A higher temporal resolution (e.g., annual) of glacial lake mapping and the expansion process of different types of glacial lakes are still limited in previous studies [34]. In this study, 46 mid- and high-resolution remote sensing images based on Landsat and Keyhole series satellite were used to provides a complete mapping (1964–2020) of glacial lakes and glaciers in the Tama Koshi (Rongxer) basin, a less reported China-Nepal transnational catchment in the central Himalayas. We aim to (1) analyze the dynamics of different types of glaciers and glacial lakes and (2) discuss the glacier-lake interactions, possible forcing mechanisms of glacial lakes expansion, their future evolution, and GLOF risk implications.

## 2. Study Area

The Tama Koshi basin (a sub-basin of the Koshi River Basin, 28°N, 86°E, Figure 1) is a long, narrow, and deep valley extending northeast-southwest, located on the south side of the central Himalayas. The total basin area is 4117 km$^2$ (about 1/3 is in China), and the elevation ranges from less than 500 m a.s.l. at Beni Ghat, where the Tama Koshi joins the Sunkoshi, to above 8000 m a.s.l. adjacent to Mt. Cho Oyu (8188 m a.s.l.). Due to the enormous elevation difference (more than 6800 m), the ecological environment in the basin is unique, complex, and diversified. Data from the European Centre for Medium-Range Weather Forecasts (ECMWF, see Section 3.4) show that the mean annual temperature of the entire basin was 5.42 °C, and the mean annual precipitation is 1380 mm, and both showed increasing trends in the past 40 years, by about 1 °C and 300 mm, respectively (Figure 1B).

Glaciers and glacial lakes are widely distributed in the upper basin. According to the Randolph Glacier Inventory (RGI) 6.0 [13], a total of 279 glaciers are developed above 4600 m a.s.l., covering a total area of 334.8 km$^2$ (8% of the entire basin). According to the latest inventory by Wang et al. [12], glacial lakes in the basin have a total number of 229 and an area of 14.6 km$^2$ in 2018, which can be classified as five major types: cirque lake (11%), end moraine-dammed lake (54%), lateral moraine-dammed lake (5%), moraine thaw lake (1%) and supraglacial lake (16%). As reported by ICIMOD in 2020 [21], 42 lakes in the Koshi basin have been identified as potentially dangerous glacial lakes (PDGL), of which 8 are located in the Tama Koshi basin. A study by Rounce et al. [35] also noted 11 very high-risk lakes in Nepal, one of them is located in this basin. Glacier and glacial lake changes in the Tama Koshi basin have been reported previously, but most with a low spatial-temporal resolution [34,36,37]. One of the most reported glacial lakes in upper Tama Koshi is the Tsho Rolpa (Rolwaling Valley), which has been evaluated as the most dangerous glacial lake in Nepal [38]. The Tsho Rolpa has been monitored, measured, and modeled since the 1990s with comprehensive details about its expansion mechanisms [39] and failure risks [40]. Various risk prevention measurements, e.g., siphon system, automatic early warning system, and artificial channel excavation, have been carried out at the Tsho Rolpa [41].

Several studies have predicted the possible response of glaciers and glacial lakes under the climate change in the Tama Koshi basin, provided the management experience and scheme of dangerous glacial lakes and discussed the potential risks of GLOFs [37,40,42,43].

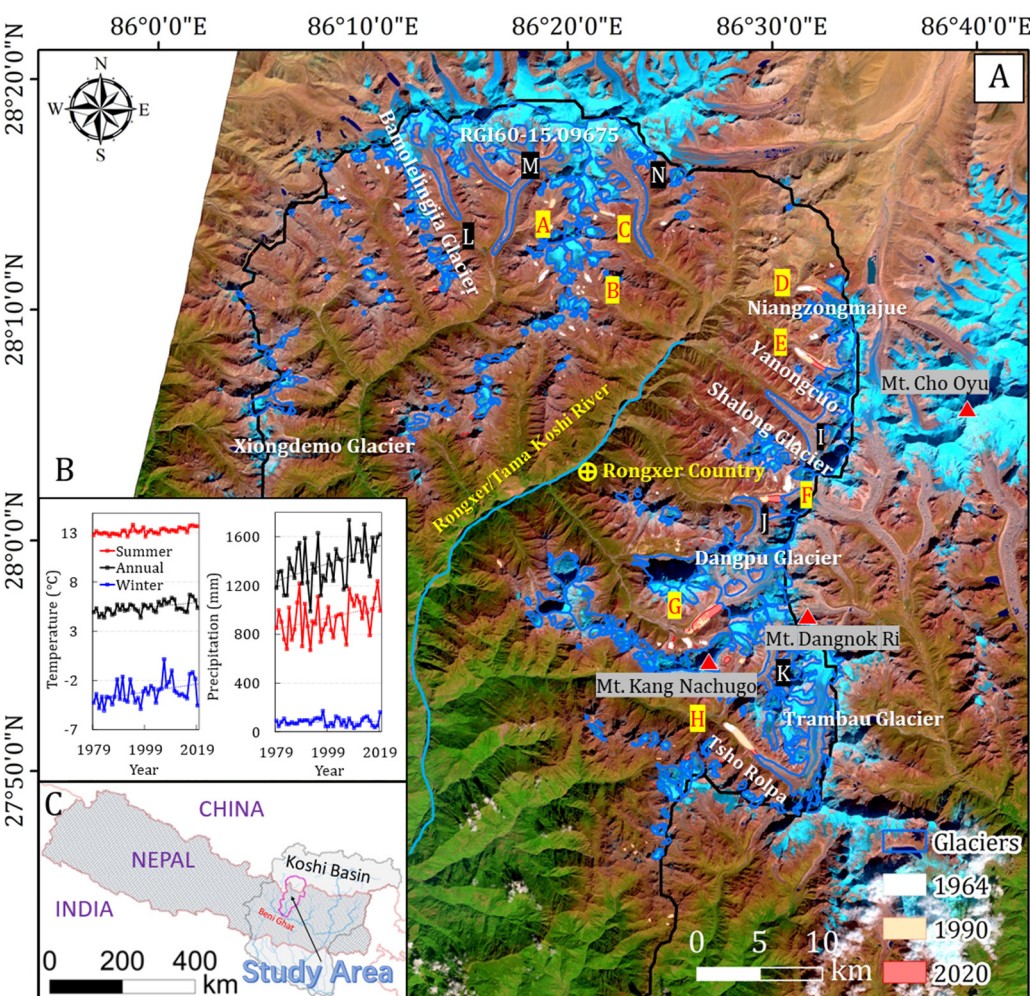

**Figure 1.** (**A**) Distribution of glaciers and glacial lakes in the Tama Koshi basin with the background image of Landsat 8 (24 October 2018), letters indicate detailly investigated glaciers and glacial lakes. The inset (**B**) shows the mean annual (red), summer (blue), and winter (black) air temperature (**left**) and precipitation (**right**) extracted from ERA5. The inset (**C**) indicates the location of the study area.

## 3. Data and Methods

### 3.1. Data Set Used

The remote sensing images used in this study were obtained from the United States Geological Survey (USGS, https://earthexplorer.usgs.gov/; accessed on 26 October 2020), Geospatial Data Cloud (http://www.gscloud.cn/; accessed on 30 October 2020) and PlanetLabs (https://www.planet.com/; accessed on 21 October 2020) between 1964 and 2020. A total of 6 KH-4A and KH-9 Keyhole (KH) images (2.7–9 m), 36 Landsat (TM/ETM+/OLI) surface reflectance images (30 m), and 4 PlanetScope (PL) Ortho Tile imagery (3.125 m) were used to manually delineated glaciers in 1964, 1980, 1990, 2000, 2010, 2018, and 2020 and glacial lakes in 1964, 1980, and automatically delineated glacial lakes in 1987–2020, respectively. Images were mainly acquired in post-monsoon and near the end of ablation seasons (August to December) with less cloud cover and seasonal snow cover. The orthorectified PL products were used as base images for the registration of KH images.

Other data sets we have used include: (1) RGI 6.0 [44], (2) Glacial Lake Inventory of High-mountain Asia in 1990 and 2018 [12], (3) ERA5 monthly aggregates from 1979 to

2019 (latest climate reanalysis produced by ECMWF/Copernicus Climate Change Service (ECCCS)), (4) the 30 m Shuttle Radar Topography Mission Digital Elevation Model (SRTM DEM), (5) 2007 ALOS PALSAR DEM (12.5 m), (6) 1975–2000, 2000–2016 glacier surface elevation change for the HMA derived by differencing KH-9 DEM (generated from optical stereo) and ASTER DEMs [18], (7) 1999–2003, 2013–2015 glacier surface flow velocity for the HMA derived by optical satellite images via feature-tracking algorithm [45], (8) Land-Scan: global population distribution data for 2000 and 2018 [46,47], and (9) 1:72,000 scale topographic map (base map from Esri). The glacier and glacial lake data sets were used to extract relevant attributes (for example, aspect and ID) and provide a reference for shaping delineations, and the DEM and glacier thickness change data were used to extract terrain parameters and glacier surface elevation changes. The attribute information and purpose of each data set are detailed in Supplementary Materials, Table S1.

### 3.2. Glacier and Glacial Lake Outline Delineation

The KH series images were characterized by high resolution, single band (panchromatic), and lack of spatial coordinates and projection information [48,49], which needs to be preprocessed before mapping. Four high-resolution orthorectified PL images (as the base image) from September to October 2020 were used for the preliminary spatial registration of KH images (as the warp image) using the ENVI Image-to-Image Registration Tool. A total of 103 (for KH-4A) and 72 (for KH-9) ground control points were selected in stable terrains such as ridgelines, river intersections, and moraine ridgelines. According to the ground control point pairs in the two images, a functional relationship is established, and the geometric correction of the distorted image is completed using coordinate transformation.

To monitor the detailed annual change of the rapidly expanding glacial lakes (Figure 1A, A–H), we automatically delineated the glacial lakes (Figure S1) in 1987–2020 (September–November) based on the Google Earth Engine (GEE) remote sensing cloud processing platform [50] Python Application Programming Interface (API) [51]. The Normalized Difference Water Index (*NDVI*; Equation (1)) and Normalized Difference Snow Index (*NDSI*; Equation (2)) were combined and applied to multi-temporal Landsat (TM/ETM+/OLI) images to reduce the errors caused by sediments and seasonally frozen lake water. Then, it is complemented with manual editing to extract glacial lake boundaries. The combination of this object-oriented image processing method and expert knowledge has shown suitable performance in glacial lake delineation in the Himalayas [52].

$$NDWI = \frac{green - NIR}{green + NIR} \tag{1}$$

$$NDSI = \frac{green - SWIR1}{green + SWIR1} \tag{2}$$

where green, *NIR*, and *SWIR*1 are the green, near-infrared, and shortwave-infrared 1 bands, respectively. By using the Python API interactive interface, the classification thresholds can be adjusted more accurately than the Javascript API interface based on the on-screen inspection of the original images. All Landsat and PlanetScope images were atmospheric and topographic corrected products. Additionally, an error of ±0.5 pixels was estimated to calculate the area uncertainties (i.e., multiply linear error and perimeter, e.g., the linear error of Landsat images are 15 m, and of KH-4A images are ~1.35 m) [33,53]. For supraglacial lakes, only the base-level lakes with perennial water storage were retained, while the perched lakes with large seasonal variations were removed [54].

### 3.3. Glacier Surface Dynamics Analysis

Eight lake-terminating glaciers (Figure 1A, A–H) and six land-terminating glaciers (Figure 1A, I–N) were selected to compare their dynamics and relationships with glacio-geomorphology or lake-ice interactions. Based on the 2000 SRTM, the two periods of glacier surface elevation in 1975 and 2016 were calculated by combining the 1975–2000 and

2000–2016 glacier surface elevation change. Surface gradient was calculated based on the 2007 ALOS PALSAR DEM using the QGIS slope analysis module, by which we extracted values between the terminal moraine to the mid-point position of the glacier tongue to better explore the slope of the downstream tongue. Subsequently, equidistant points at a 30 m spacing were generated along the entire glacier center flow line to identify and visualize the elevation, flow velocity, and slope. For the elevation and velocity analysis of the ice/lake area, we extended the line to a further 300–800 m in front of the moraine dam.

### 3.4. Meteorological Data Extractions

Since there is no meteorological observation in the Tama Koshi basin, we used the ERA5 monthly aggregates data to analyze the impact of meteorological factors. The ERA5 data set, with a spatial resolution of 0.25 radian, is the fifth generation ECMWF atmospheric reanalysis of the global climate [55]. Its grided data set is now incorporated in the GEE data catalog, and we extracted the temperature and precipitation data inside the basin domain between 1979 and 2019 pixel by pixel.

## 4. Results

### 4.1. Glacier Distribution and Changes

#### 4.1.1. Distribution of Glaciers

Our glacier inventories show (Figure 2A) a total of 271 glaciers with a total area of $329.2 \pm 1.9$ km$^2$ in 2020 (Table 1). The largest one is Glacier M ($42.6 \pm 0.1$ km$^2$), which is a debris-covered glacier facing south and located in the upper Lapche, one of the western tributaries of the Tama Koshi (Figure 1, Randolph Glacier Inventory ID: RGI60-15.09675). The mean elevation of all glaciers ranges from 4760 to 6270 m a.s.l. (Figure 2B) and most (96%) glacier areas are distributed between 5200 and 5800 m a.s.l. The lowest glacier tongue, which is close to the Rongxer country (ID: RGI60-15.09760), extends to an altitude of 4475 m a.s.l. Due to the northeast (high altitude) to the southwest (low altitude) orientation of the basin, southern oriented glaciers have the maximum area coverage ($84.9 \pm 0.4$ km$^2$, ~26%, Figure 2C). Northwest-oriented glaciers have the second largest area of $68.2 \pm 0.3$ km$^2$ (~21%), whereas the glaciers with the east orientation are the smallest ($10.3 \pm 0.1$ km$^2$, 3.1%).

**Table 1.** Glacier inventories between 1964 and 2020 of the Tama Koshi basin.

| Study Period (Sensor) | Glacier Number | Total Area (km$^2$) | Area Change (% a$^{-1}$) | West Side of Basin | | East Side of Basin | |
|---|---|---|---|---|---|---|---|
| | | | | Number | Area (km$^2$) | Number | Area (km$^2$) |
| 1964 (KH-4A) | 282 | $355.3 \pm 1.8$ | - | 141 | $142.0 \pm 0.8$ | 141 | $213.4 \pm 1.0$ |
| 1980 (KH-9) | 283 | $350.4 \pm 5.9$ | $-0.09$ | 143 | $141.2 \pm 2.6$ | 140 | $209.3 \pm 3.4$ |
| 1990 (Landsat TM) | 279 | $337.5 \pm 19.0$ | $-0.37$ | 139 | $136.2 \pm 8.1$ | 140 | $201.3 \pm 10.9$ |
| 2000 (Landsat TM) | 279 | $335.9 \pm 19.0$ | $-0.05$ | 139 | $135.5 \pm 8.1$ | 140 | $200.5 \pm 10.9$ |
| 2010 (Landsat ETM+) | 279 | $335.0 \pm 18.9$ | $-0.03$ | 139 | $135.2 \pm 8.1$ | 140 | $199.8 \pm 10.8$ |
| 2018 (Landsat OLI) | 271 | $331.2 \pm 18.5$ | $-0.14$ | 133 | $133.9 \pm 7.9$ | 138 | $197.4 \pm 10.6$ |
| 2020 (PlanetScope) | 271 | $329.2 \pm 1.9$ | $-0.31$ | 133 | $133.8 \pm 0.8$ | 138 | $195.4 \pm 1.1$ |

Glacier distributions are slightly different between the basin's east and west sides, divided by the Rongxer/Tama River corridor (Figure 1 and Table 1). On the west side of the corridor, the three largest debris-covered glaciers (L, M, and N in Figure 1) occupied ~63% of the total glacier area (133.7 km$^2$) compared to the other 131 relatively smaller glaciers (50.0 km$^2$). In contrast, mean glacier areas in the east (1.4 km$^2$ for the 138 glaciers) are larger than the west (1.0 km$^2$), with seven glaciers are >5 km$^2$ and five are >10 km$^2$. Glaciers on the east side distribution with a wide altitude range (mainly between 5200 and 5800 m a.s.l.), whereas the west side glacier areas are concentrated between 5600 and 5700 m a.s.l. All five lake-terminating glaciers are located on the east side, but the supraglacial lakes are more frequently found on the lower part of the three largest debris-covered glaciers.

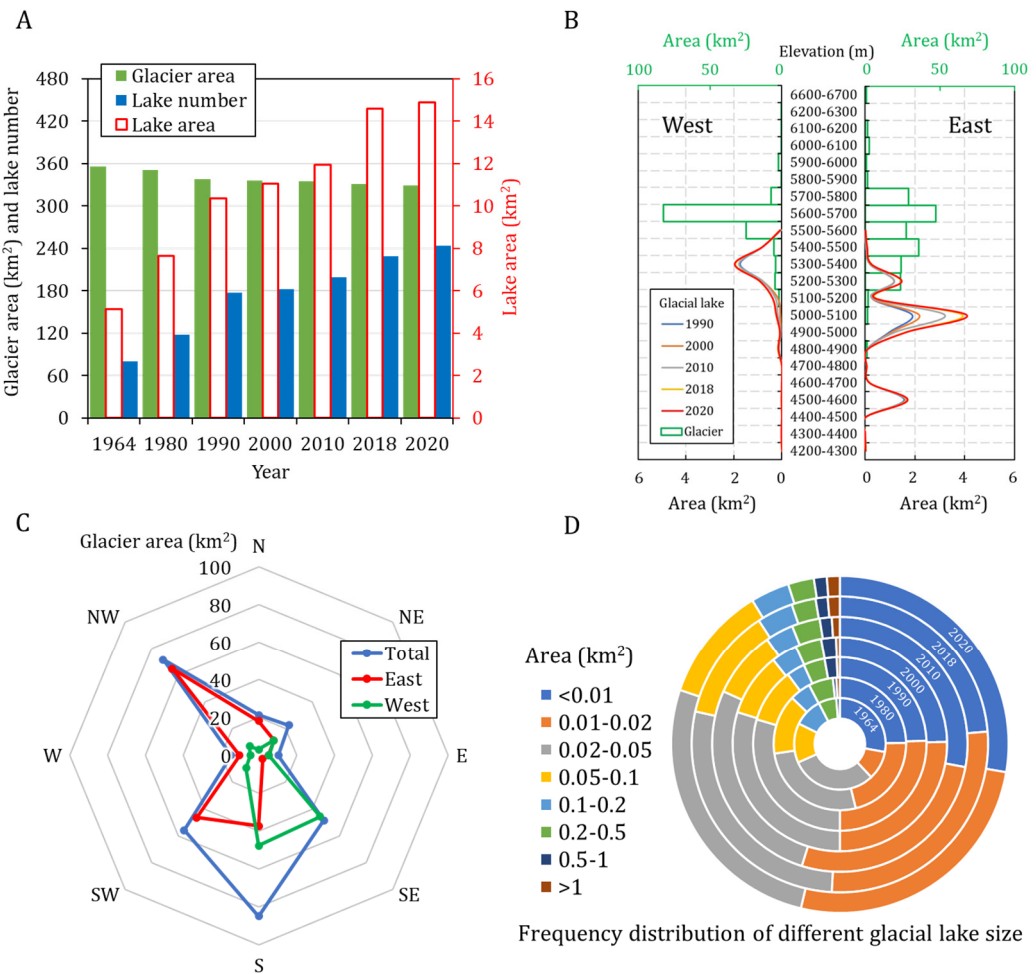

**Figure 2.** Overview of glaciers and glacial lakes. (**A**) Number and area changes of glaciers and glacial lakes from 1964 to 2020. (**B**) Elevation distribution of glaciers (in 2020) and glacial lakes (between 1990 and 2020) on the west and east side of the basin. (**C**) Glacier distribution with aspect in the total (blue), east (red), and west (green) of the basin. (**D**) Frequency of glacial lakes in various size classes.

### 4.1.2. Glacier Changes

From 1964 to 2020, the total area of glaciers in the Tama Koshi basin decreased by $26.2 \pm 3.2$ km$^2$ (0.13% a$^{-1}$) in 56 years (Tables 1 and 2), with a mean glacier terminus retreat rate of 3.2 m a$^{-1}$. Glacier area loss rates in the periods of 1980–1990 and 2018–2020 were 0.37% a$^{-1}$ and 0.31% a$^{-1}$, respectively, which were much higher than the rest study period (0.03–0.14% a$^{-1}$, Table 1). Eight small glaciers covering an area of 0.03–0.29 km$^2$ had disappeared between 1980 and 1990. The mean retreat rate of lake-terminating glaciers (37.7 m a$^{-1}$) is about 14 times higher than those land-terminating ones (2.6 m a$^{-1}$). Glacier G, which terminates in lake G, experienced the longest terminus retreat (68 m a$^{-1}$) between 1964 and 2020. On the east side of the basin, glacier area loss rate (0.15% a$^{-1}$) and retreat rate (4.3 m a$^{-1}$) are generally higher than those on the west (0.10% a$^{-1}$ and 2.08 m a$^{-1}$, respectively, Table 2).

**Table 2.** Glacier changes between 1964 and 2020 for different glacier types and locations.

| | Study Period | Total | Land-Terminating | Lake-Terminating | West Side of Basin | East Side of Basin |
|---|---|---|---|---|---|---|
| Area change rate (% a$^{-1}$) | 1964–2020 | 0.13 | 0.13 | 0.14 | 0.10 | 0.15 |
| Retreat rate (m a$^{-1}$) | 1964–2020 | 3.2 | 2.6 | 37.7 | 2.1 | 4.3 |
| Elevation change rate (m a$^{-1}$) | 1975–2000 | −0.32 | −0.30 | −0.40 | −0.26 | −0.33 |
| | 2000–2016 | −0.63 | −0.62 | −0.85 | −0.69 | −0.62 |
| Mean Velocity (m a$^{-1}$) | 1999–2003 | 5.3 | 4.8 | 6.9 | 5.4 | 5.2 |
| | 2013–2015 | 4.0 | 3.4 | 6.1 | 4.1 | 3.8 |

Note: From 1975 to 2000, glacier-wide mean elevation changes were extracted using the glacier boundary in 1980 and from 2000 to 2016 using the boundary in 2000. From 1999 to 2003, mean velocities were extracted using glacier boundaries in 2000 and from 2013 to 2015 using glacier boundaries in 2010.

Meanwhile, glaciers also showed a trend of thinning and slowdown (Table 2). The mean change rates of glacier surface elevation during 1975–2000 and 2000–2016 were −0.32 m a$^{-1}$ and −0.63 m a$^{-1}$, respectively, indicating an accelerated thinning of the ice. Glacier elevation change rates on the western side (from −0.26 m a$^{-1}$ in the period 1975–2000 to -0.69 m a$^{-1}$ in the period 2000–2016) shows more remarkable than the east (−0.33 m a$^{-1}$ in the period 1975–2000 to 0.62 m a$^{-1}$ in the period 2000–2016) (Table 2). The most downwasted glacier on the west side of the basin is the Glacier L, one of the largest land-terminating glaciers (debris-covered, with a length of 12.7 km) in the upper Tama Koshi. Its mean elevation change rate increased from −0.45 m a$^{-1}$ in 1975–2000 to −0.97 m a$^{-1}$ in 2000–2016 (Figure 3), higher than that of Glacier E, I, and G on the east side (−0.78 m a$^{-1}$, −0.60 m a$^{-1}$ and −0.89 m a$^{-1}$ in 2000–2016, respectively). Nevertheless, the mean elevation change rates of lake-terminating glaciers (−0.40 m a$^{-1}$ in the period 1975–2000 and −0.85 m a$^{-1}$ in the period 2000–2016) in the basin were higher than that of land-terminating glaciers (−0.30 m a$^{-1}$ in the period 1975–2000 and −0.62 m a$^{-1}$ in the period 2000–2016) during the two study periods.

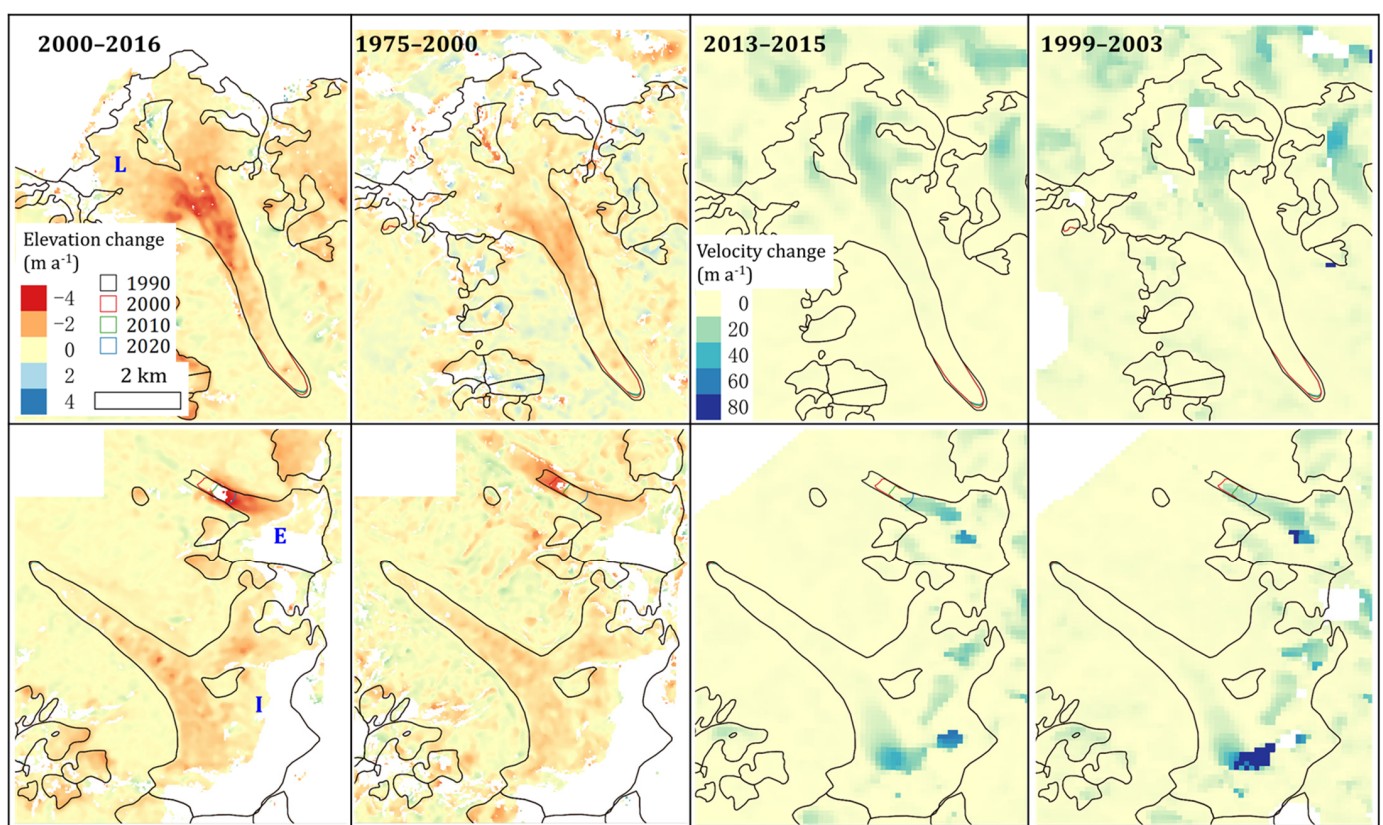

**Figure 3.** Comparisons of elevation change rates and velocities of three typical land- and lake-terminating glaciers (Glacier E, I, and L) on the east and west sides of the basin.

The mean glacier surface velocity of the Tama Koshi basin decreased by ~25% from 5.3 m a$^{-1}$ in 1999–2003 to 4.0 m a$^{-1}$ in 2013–2015. Overall, mean glacier surface velocities on the west side (5.4 m a$^{-1}$ in the period 1999–2003 and 4.1 m a$^{-1}$ in the period 2013–2015) were slightly higher than on the east side (5.2 m a$^{-1}$ in the period 1999–2003 and 3.8 m a$^{-1}$ in the period 2013–2015), while both have decelerated during the two study periods. Lake-terminating glaciers (with a velocity of 6.9 m a$^{-1}$ in the period 1999–2003 and 6.1 m a$^{-1}$ in the period 2013–2015) moved faster than land-terminating glaciers (4.8 m a$^{-1}$ in the period 1999–2003 and 3.4 m a$^{-1}$ in the period 2013–2015). Even though the surface velocity declined by 11%, it was much slower than that of land-terminating glaciers (−29%).

### 4.2. Glacial Lake Distribution and Changes

#### 4.2.1. Distribution of Glacial Lakes

The statistical results of glacial lake inventories between 1964 and 2020 are presented in Table 3. A total of 196 glacial lakes (more than 50% are moraine-dammed proglacial lakes) are developed in the basin, with a total area of 14.4 ± 0.3 km$^2$ and a mean area of 0.07 km$^2$ in 2020. These values are slightly lower than the glacial lake inventory in 2018 (see Section 2) published by Wang [12] but much higher than Wu's study in 2005 (lake number: 28; lake area: 7.87 km$^2$) [34]. There are 107 glacial lakes (~55% of the total number of glacial lakes) with an area of >0.02 km$^2$ (Figure 2D), which occupy 13.3 km$^2$ (~93%) of the total lake area. Most glacial lakes (14.2 ± 0.3 km$^2$, ~99%) are distributed between 4500 and 5500 m a.s.l. (Figure 2B) and most are located on the east side of the corridor. The total area of the glacial lake on the east side (10.0 ± 0.1 km$^2$, ~69%) is much larger than that of the west (4.4 ± 0.1 km$^2$). The altitudes of the glacial lakes on the west side (5100–5500 m a.s.l.) are generally higher than the east (4500–5300 m a.s.l.). Nevertheless, there are a total of 119 glacial lakes on the western side, accounting for ~61% of the total number of our mapped glacial lakes. However, all of them are less than 0.3 km$^2$ in area.

**Table 3.** Glacial lake inventories between 1964 and 2020 and changes in meteorological conditions in the Tama Koshi basin.

| Study Period (Sensor) | Lake Number | Total Area (km$^2$) | Area Change (% a$^{-1}$) | West Side of Basin | | East Side of Basin | | Temperature Change (°C a$^{-1}$) | Precipitation Change (mm a$^{-1}$) |
|---|---|---|---|---|---|---|---|---|---|
| | | | | Number | Area (km$^2$) | Number | Area (km$^2$) | | |
| 1964 (KH-4A) | 78 | 5.1 ± 0.1 | - | 51 | 2.0 ± 0.1 | 27 | 3.1 ± 0.1 | - | - |
| 1980 (KH-9) | 115 | 7.6 ± 0.5 | 3.0 | 76 | 3.0 ± 0.6 | 39 | 4.5 ± 0.2 | - | - |
| 1990 (Landsat TM) | 171 | 10.3 ± 2.2 | 3.6 | 108 | 3.8 ± 1.1 | 63 | 6.5 ± 1.1 | - | - |
| 2000 (Landsat TM) | 172 | 10.7 ± 2.2 | 0.47 | 109 | 4.0 ± 1.1 | 63 | 6.9 ± 1.1 | 0.033 | 0.48 |
| 2010 (Landsat ETM+) | 175 | 11.7 ± 2.3 | 0.85 | 112 | 3.8 ± 1.1 | 63 | 7.8 ± 1.1 | 0.038 | 6.4 |
| 2018 (Landsat OLI) | 198 | 14.2 ± 2.7 | 2.8 | 118 | 4.4 ± 1.3 | 80 | 9.8 ± 1.4 | 0.0007 | 14.6 |
| 2020 (PlanetScope) | 196 | 14.4 ± 0.3 | 0.49 | 119 | 4.4 ± 0.1 | 77 | 10.0 ± 0.1 | - | - |
| Total | | | | | | | | | |
| Lake growth (dA%) | 151 | - | | 180 | 133 | 123 | 185 | 217 | - | - |
| Lake growth rate (% a$^{-1}$) | 2.7 | - | | 3.2 | 2.4 | 2.2 | 3.3 | 3.9 | - | - |

Note: Temperature and precipitation changes were obtained by calculating the difference between ten-year averages.

#### 4.2.2. Glacial Lake Changes

From 1964 to 2020, the total number and area of glacial lakes in the Tama Koshi basin increased considerably (by ~151% in number and ~180% or 3.2% a$^{-1}$ in area, Figure 2A). Lake expansion between 1980 and 1990 reached the highest rate of 3.5% a$^{-1}$ (Table 3). Statistically, a total of 12 glacial lakes disappeared during 2000–2020, 36 glacial lakes were newly formed, 10 glacial lakes shrank (7 in the west), and 26 glacial lakes expanded. The glacial lakes that develop between 5000 and 5100 m a.s.l. have experienced the highest expansion rate, with a rate of 4.0% a$^{-1}$.

In 1964, there were only 27 glacial lakes on the eastern side of the basin, with a total area of 3.1 ± 0.1 km$^2$ (Table 3). This number in 2020 has increased to 77 by a rate of 3.3% a$^{-1}$ (185%), which is higher than the 2.4% a$^{-1}$ (133%) on the western side (from 51 in 1964 to 119 in 2020, Table 3). Their overall expansion rate (3.9% a$^{-1}$) was much higher than the west side (2.2% a$^{-1}$). According to Table 4, moraine-dammed ice-contacted

glacial lakes show a much higher expansion rate (~204%) than non-moraine-dammed ice-contacted lakes (~175%). According to our multi-decade inventories (Table 4), the total number of moraine-dammed ice-contacted lakes was peaked around 1980 at 17, an increase from 10 in 1964. After 1980, the number of moraine-dammed ice-contacted glacial lakes gradually decreased to five in 2020; the other 12 were disconnected from the retreating mother glacier. The annual expansion rates during periods after 1980 ($0.4\%$ $a^{-1}$–$2.2\%$ $a^{-1}$) were also lower than the period before1980 ($5.7\%$ $a^{-1}$, 1964–1980). In contrast, for the non-ice-contacted lakes, their numbers showed a continuous increase from 68 in 1964 to 191 in 2020, while their mean annual area expansion rate ($3.1\%$ $a^{-1}$) is close to those moraine-dammed ice-contacted lakes ($3.7\%$ $a^{-1}$).

**Table 4.** The expansion differences between ice-contact lake and non-ice-contact lake in the Tama Koshi basin.

| Study Period (Sensor) | Moraine-Dammed Ice-Contact Lake | | | Non-Moraine-Dammed Ice-Contact Lake | | |
|---|---|---|---|---|---|---|
| | Number | Area (km$^2$) | Expansion Rate (% a$^{-1}$) | Number | Area (km$^2$) | Expansion Rate (% a$^{-1}$) |
| 1964 (KH-4A) | 10 | $2.1 \pm 0.1$ | - | 68 | $3.0 \pm 0.1$ | - |
| 1980 (KH-9) | 17 | $3.8 \pm 0.2$ | 5.7 | 98 | $3.7 \pm 0.3$ | 1.5 |
| 1990 (Landsat TM) | 11 | $4.0 \pm 0.4$ | 0.40 | 160 | $6.3 \pm 1.8$ | 6.9 |
| 2000 (Landsat TM) | 12 | $4.9 \pm 0.6$ | 2.2 | 160 | $5.9 \pm 1.7$ | −0.65 |
| 2010 (Landsat ETM+) | 10 | $5.3 \pm 0.5$ | 0.83 | 165 | $6.4 \pm 1.9$ | 0.85 |
| 2018 (Landsat OLI) | 5 | $5.8 \pm 0.5$ | 1.3 | 193 | $8.4 \pm 2.4$ | 4.0 |
| 2020 (PlanetScope) | 5 | $6.1 \pm 0.1$ | 2.2 | 191 | $8.3 \pm 0.3$ | −0.71 |
| Total | | | | | | |
| Lake growth (dA%) | −50 | - | 204 | 181 | - | 175 |
| Lake growth rate (% a$^{-1}$) | −0.9 | - | 3.7 | 3.2 | - | 3.1 |

Between 1964 and 2020, the lake size distribution has also varied with time (Figure 2D). Glacial lakes with an area of <0.01 km$^2$ and between 0.05 and 0.2 km$^2$ showed the most unstable, while lake sizes between 0.01 and 0.05 km$^2$ gradually increased. For those glacial lakes with larger sizes, e.g., lakes with a size of 0.2–0.5 km$^2$ were relatively stable, and lake size of >1 km$^2$ showing more expansive during 1964–1990 following by relatively stable after 1990.

In this study, we found that two types of glacial lakes have experienced rapid expansion: moraine-dammed ice-contact lake and supraglacial lake (Figures 4 and 5, and Table 4). Based on the results of potentially dangerous glacial lakes evaluation from ICIMOD [21], the actual changes of the glacial lakes, and the past GLOF events (see Section 5.4), we identified eight (seven (B-H) of them are same with ICIMOD's report) rapidly expanding moraine-dammed ice-contact lakes (Lake A–H, Figures 1 and 6) as PDGLs. Lake G is a moraine-dammed lake formed around in 2009 by the expanding and coalescing with several supraglacial ponds (most of which are base-level lakes) in the 1980s. The total lake area over the entire glacier basin has increased by $9.0\%$ $a^{-1}$ from 0.08 km$^2$ in 1980 to 1.92 km$^2$ in 2020. Lakes A–C, located on the west side of the basin, expanded rapidly before 1990 but then gradually stabilized in their size. Lake A has shrunk significantly between 2018 and 2020, with a rate of ~−36% (or 0.08 km$^2$ $a^{-1}$). It was confirmed that the lake burst in 2018 according to the comparison of satellite images (more details are given in Section 5.4 and Figure S2). The expansion rate of Lake D (Niangzongmajue) was maintained at $1.3\%$ $a^{-1}$ before 2014 and then remained relatively stable, with no evident expansion in the following six years. Lake E (Yalongcuo) has experienced steady growth ($1.9\%$ $a^{-1}$) over the past 56 years. The expansion rate of Lake F increased rapidly after 2004, from $3.8\%$ $a^{-1}$ between 1990 and 2004 to $5.8\%$ $a^{-1}$ between 2004 and 2020, with an overall area growth of 52.9%. Lake H (Tsho Rolpa) expanded clearly before 1990 at a rate of ~$3.5\%$ $a^{-1}$, and then its area remained virtually unchanged after the manual intervention (drainage to lower down the lake level) in the 1990s. However, after 2012, despite continued intervention, it still showed

a slight expansion trend with a rate of 0.8% a$^{-1}$. The above results are summarized in Table 5.

**Table 5.** Rates of changes and other related information for several lake-terminating glaciers (Glacier A-H) and land-terminating glaciers (Glacier I-N).

| ID | Name (Lake or Glacier ID) | (1964–2020) Rate of Change in Lake Area (% a$^{-1}$) | (1999–2015) Rate of Change in Glacier Velocity (m a$^{-1}$) | (1975–2016) Change in Glacier Elevation (m a$^{-1}$) | (2007) Average Slope (°) |
|----|---------------------------|------|------|------|------|
| A | RGI60-15.09690 | −1.9 | 2.1 | −0.55 | 5.5 |
| B | RGI60-15.09714 | 0.33 | 4.8 | −0.55 | 5.8 |
| C | RGI60-15.09721 | 0.84 | 9.5 | −0.49 | 5.2 |
| D | Niangzongmajue | 1.1 | 25.4 | −0.40 | 5.5 |
| E | Yalongcuo | 1.9 | 10.4 | −0.80 | 3.1 |
| F | Dangpu Lake | 4.2 | 2.8 | −1.2 | 7.7 |
| G | RGI60-15.09771 | 9.0 | 7.4 | −1.1 | 5.5 |
| H | Tsho Rolpa | 0.60 | 9.7 | −0.71 | 3.0 |
| I | Shalong Glacier | 0.04 | 5.1 | −0.64 | 7.9 |
| J | Dangpu Glacier | - | 3.7 | −0.47 | 10.5 |
| K | RGI60-15.03428 | - | 2.5 | −0.71 | 7.9 |
| L | Bamolelingjia Glacier | - | 5.7 | −0.86 | 8.2 |
| M | RGI60-15.09675 | - | 8.9 | −0.64 | 7.8 |
| N | RGI60-15.09729 | - | 14.7 | −0.26 | 8.1 |

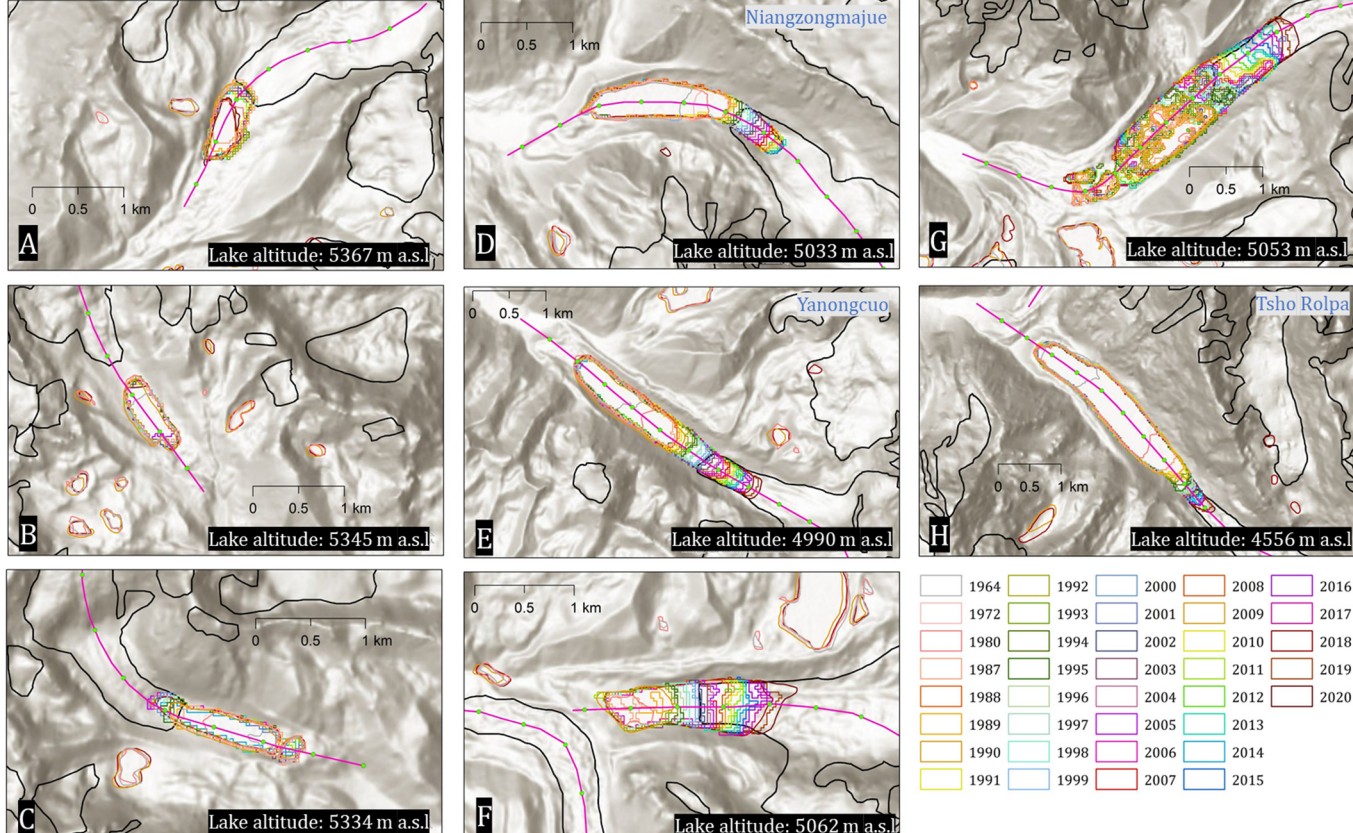

**Figure 4.** Changes of eight (Lake **A–H**) rapidly expanding proglacial lakes in the past 56 years overlayed on the Esri topographic map.

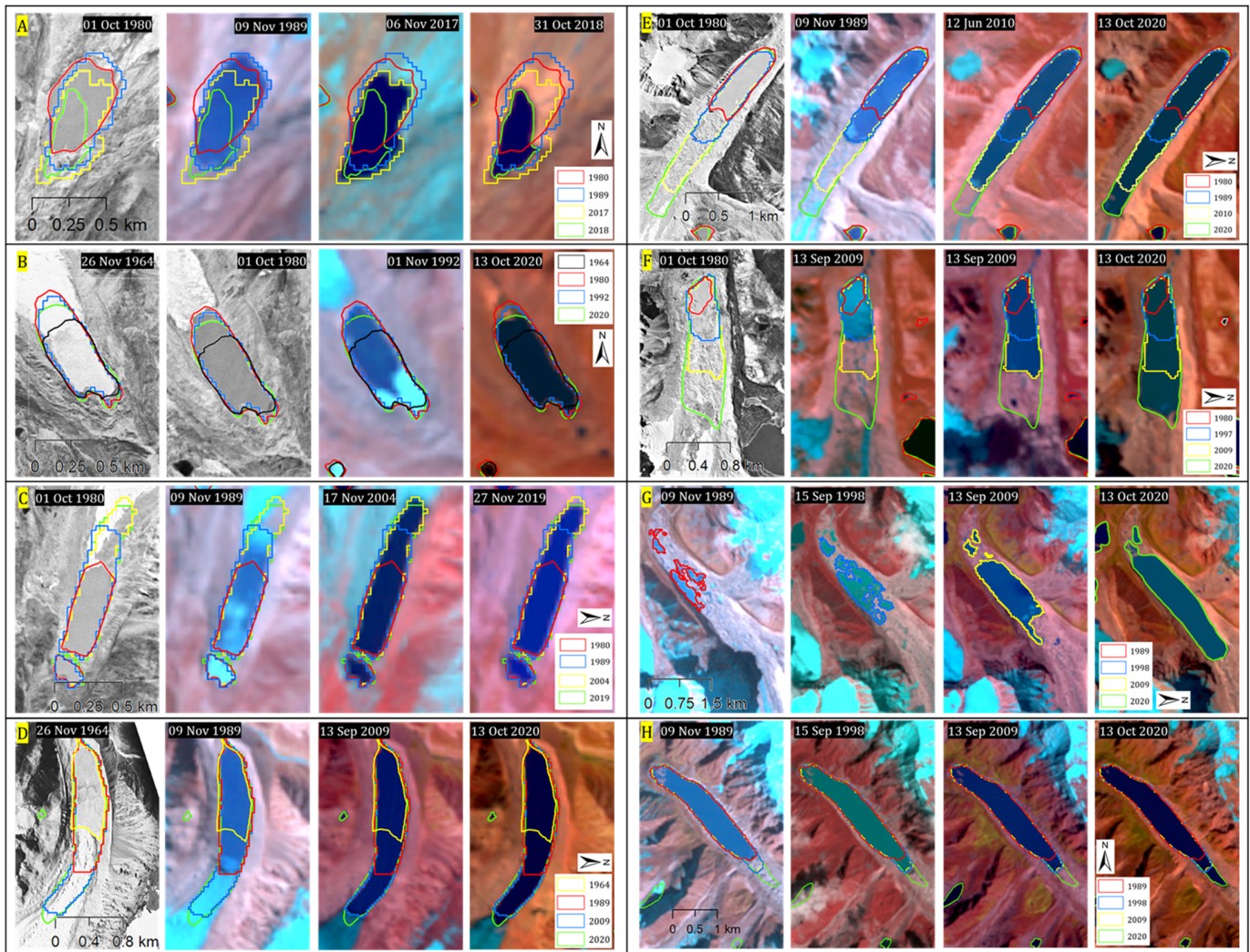

**Figure 5.** The expansion process of eight (Lake **A–H**) rapidly expanding proglacial lakes with the background of KH and Landsat images.

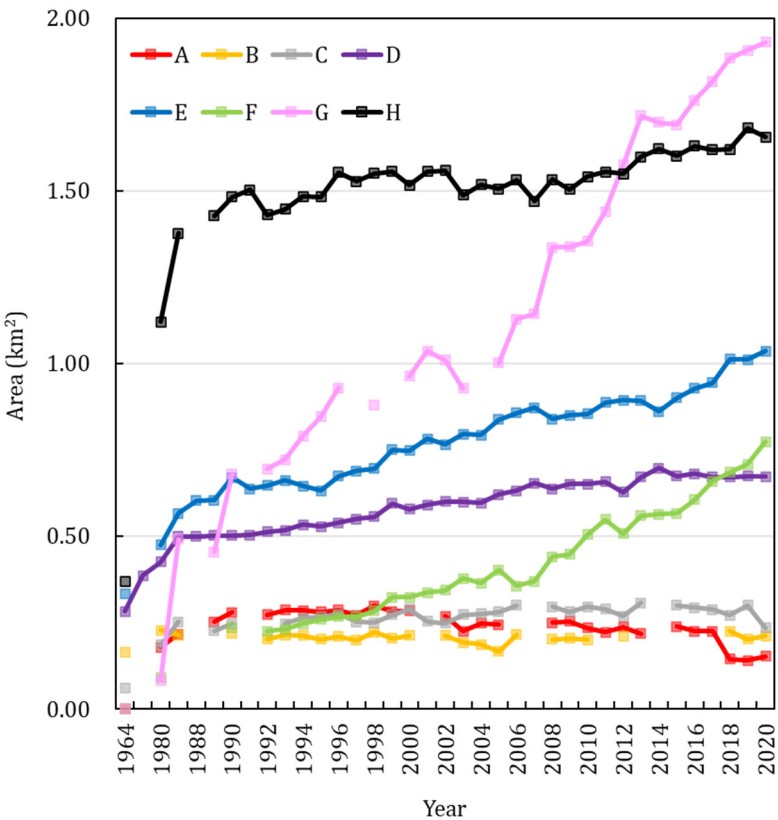

**Figure 6.** Area changes of eight (Lake **A**–**H**) rapidly expanding proglacial lakes.

## 5. Discussion

### 5.1. Glacier and Glacial Lake Variations and Climate Changes

As most HMA lakes are marginally influenced by direct human activities, glacial lake variations are closely related to climate change. Air temperature increase is considered to be the main driving force for glacier retreat and wasting [56]. Previous studies have shown significant warming in the Himalayas. From 1979 to 2019, the temperature rising rate in the Tama Koshi basin was measured at 0.01 °C a$^{-1}$, and for winter and summer is 0.002 °C a$^{-1}$ and 0.02 °C a$^{-1}$, respectively, according to the analysis of ERA5. The precipitation changes during the past decades in the entire Himalayas were not uniform [57]. For the Tama Koshi basin, the basin-scale annual precipitation between 1979 and 2019 showed an increasing trend with a rate of 11.0 mm a$^{-1}$, and during the winter and summer seasons are 1.8 and 3.4 mm a$^{-1}$, respectively. This trend was similar to the meteorological observations at a nearby Tingri Station [11]. Although both temperature and humidity will affect evapotranspiration of the lake, the increase in temperature and precipitation has elevated freezing levels in the post-monsoon and early winter periods [58] and may boost the water supply of glacial lakes due to increased glacier ablation and runoff, which cause a negative effect on evapotranspiration [59], and eventually cause the lake level to rise (for example Lake Tsho Rolpa [60]) or induce a GLOF (please see Lake A in Section 5.4).

### 5.2. Glacier-Lake Interactions

Compared to the land-terminating glaciers, the mean change rates of area, length, elevation, and velocity of lake-terminating glaciers are remarkably higher (Table 2). In the Tama Koshi basin, length changes of most land-terminating glaciers (mostly are debris-covered) were not obvious during the study period (e.g., Glacier I, J, K, on the eastern side and Glacier L, M, N on the western side). The decrease in ice thickness, on the other hand, was their main result of mass loss. This is contrasted with those rapid retreats of lake-terminating glaciers (Figure 3). For example, for those rapidly expanding moraine-dammed

lakes (Figure 5), they are all currently connecting with glaciers or have been in contact with ice. Such significant differences in glacial retreating rates were widely observed in the HMA regions [25,61,62] since the lake-calving is suggested as one of the important mass loss processes of the lake-terminating glaciers [63,64]. When those proglacial lakes finished their rapid expansion stage and gradually separated from the mother glacier, their expansion rates become gradually decreased (e.g., Glacier A, B, C). However, the retreat rates of the ice tongue are still faster than those initial land-terminating glaciers.

In addition, glaciers terminating lakes also showed notable changes in their thicknesses and flow velocities. The surface elevations of the glaciers that are connecting with Lakes A, B, E, F, G, and H have shown notably higher lowering rates than those land-terminating debris-covered glaciers (Tables 2 and 5, e.g., I, J, K). The largest elevation change occurred on the lake-terminating glacier connected with Lake F, reaching $-1.2$ m a$^{-1}$ between 1975 and 2016 (Table 5). The remarkable high lowering rate ($-0.64$ m a$^{-1}$, Table 5) occurred at the largest debris-covered land-terminating glacier (M) is likely caused by the high-density distribution of supraglacial lakes (most of them are perched lakes) and ice cliffs (Figure S3) on its lower tongue, as these features acting as ablation hot spots on debris-covered glaciers [65–67] are unstable and seasonal. Although both lake- and land-terminating glaciers have shown a slowdown of the velocities since 1999, the decelerating rates of lake-terminating glaciers are much lower than the land-terminating glaciers (Table 2), indicating the modulation effect of proglacial calving processes on the maintenance of high ice flow rate of the lake-terminating glaciers [61].

### 5.3. Evolution and Future Development of Glacial Lakes Controlled by Topography

In general, the topography where the glacial lake develops (the pink rectangle highlighted area in Figure 7) generally is relatively flat. The mean slope of these lake-terminating glaciers is around 5°, while the mother glacier of Lake E has the lowest slope (~3°), and the mother glacier of Lake F has the highest surface gradient but still no more than 8° (Table 5). When the valley between receding glacier fronts and terminal moraines provides a sufficient and achievable area for glacial lake development, e.g., Lake D–H, the lake could continually expand. However, when the glacier gradually retreats to the region with higher elevation and steeper slope (Table 5), the expansion rate of the glacial lake gradually slows down, e.g., lake A–C has been fully loaded (or have reached their maximum upper limit). For Lake F and G, although they have high-gradient ice tongues, they are surviving thanks to their moraine dams supported by adjacent glacier-moraine complexes (Figure 4). From the longitudinal profile analysis (Figure 7), the height of the moraine dam in front of Lake F is the highest (50 m), which offers a strong bearing capacity for lake growth. On the other hand, despite the evident mass loss (with the mean glacier surface elevation change rate of $-0.6$ m a$^{-1}$), the land-terminating glaciers (I–N) are still difficult to store water due to the lack of stable dams (Figure 5) or gentle slopes (with the mean slope of 8.4°; Table 5). Supraglacial lakes on these debris-covered glaciers generally are perched lakes thus could only be filled seasonally [54], or englacial or subglacial drainage system in the glacier maybe exist that could drain the lakes periodically [68]. The number and area of supraglacial ponds on Glacier M and N are increasing slightly and may merge into a large supraglacial lake or moraine-dammed lake in the future once those perched lakes are located in the lowest part of the glacier change to base-level lakes as the glacier downwasting. Additionally, some cirque lakes may have formed after small cirque glaciers seriously downwasted or disappeared. We therefore argue that topography is not only an essential factor affecting the expansion of glacial lakes, but also a major condition for the development of glacial lakes. Moreover, the altitude and orientation of the proglacial lake may also affect the thermal condition and water balance of the glacial lake by influencing the absorption of solar radiation and ablation processes of adjacent ice.

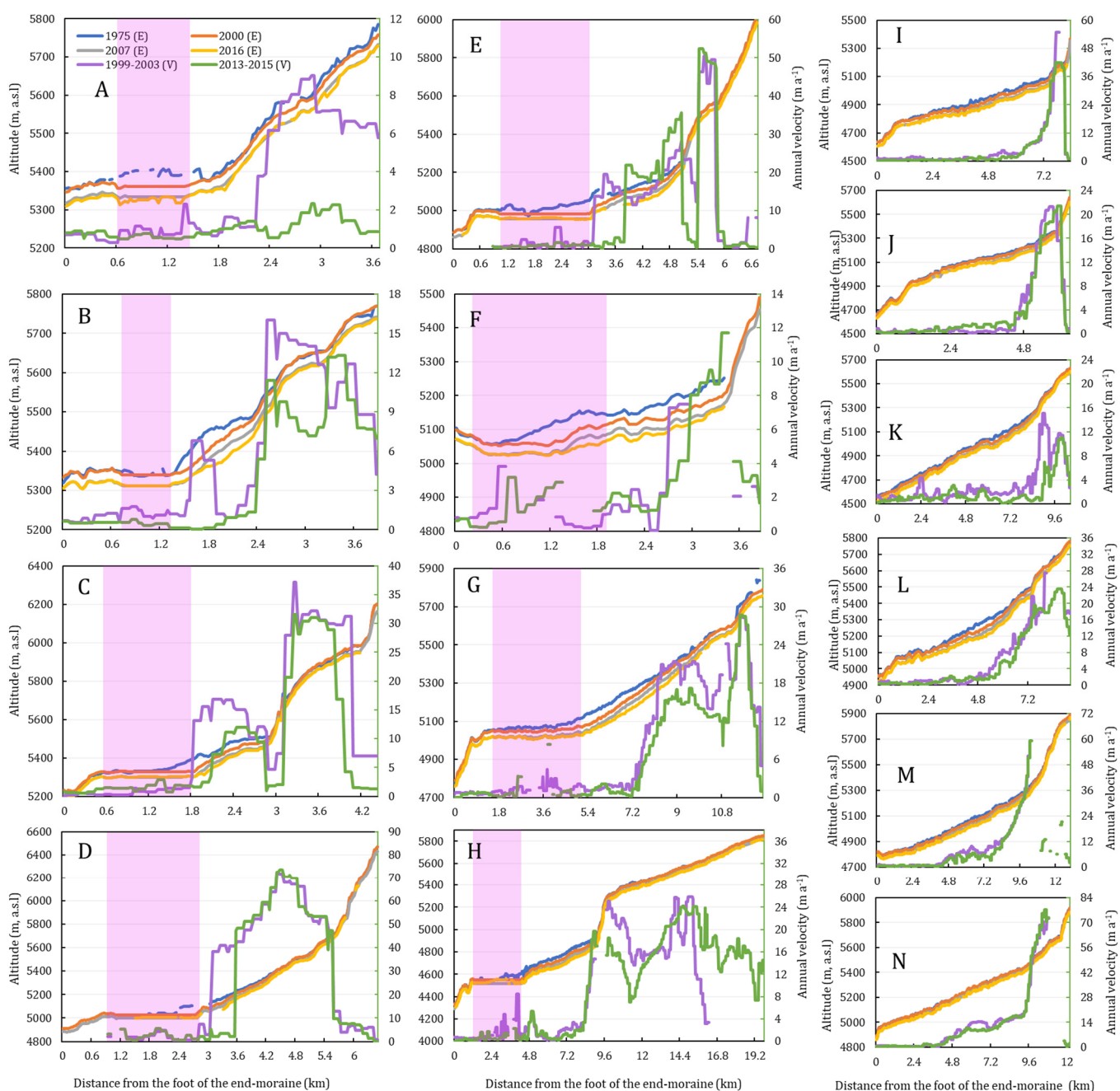

**Figure 7.** Changes in elevation and velocity of the lake-terminating glaciers (**A–H**) and land-terminating glaciers (**I–N**). "E" is elevation, "V" is velocity, the pink rectangle highlighted is the area where the glacial lake develops.

*5.4. Implications for the Risk of GLOFs*

Over 50 GLOF events have been documented in the Himalayas due to glaciers recession [69], 3 of which are located in the Tama Koshi basin (Figure 8, Table 6). The first one is entitled Chubung GLOF (86°27′38″E, 27°52′37″N), located in the Rolwaling Valley in Nepal, in front of Rolpa Tsho (true northwest). This GLOF was triggered by an ice avalanche that fell into the lake on 22 July 1991 and caused the lake to completely drain [38]. The second GLOF event (86°26′49″E, 27°55′46″N), which is named Upper Langbu Tsho, is located in the upper Dagazhuoma River, Rongxer country, Tibet, China, near Lake G (true right). The outburst of this lake from September to November 1992 was possibly triggered by an ice avalanche [69]). The third GLOF event was identified in this study by a continuous satellite tracking of Lake A (86°19′16″E, 27°14′43″N, Figure S2), which is

located in the upper Labujikongzangbu River. Remote sensing monitoring showed that Lake A was covered by thick clouds for 13 days (28 July–9 August 2018) before its failure. Meteorological results also showed that the summer precipitation in 2018 was remarkably higher than in other years (Figure 1B, with a total of 1234 mm between June and August). We speculate that the high level of precipitation triggered this GLOF.

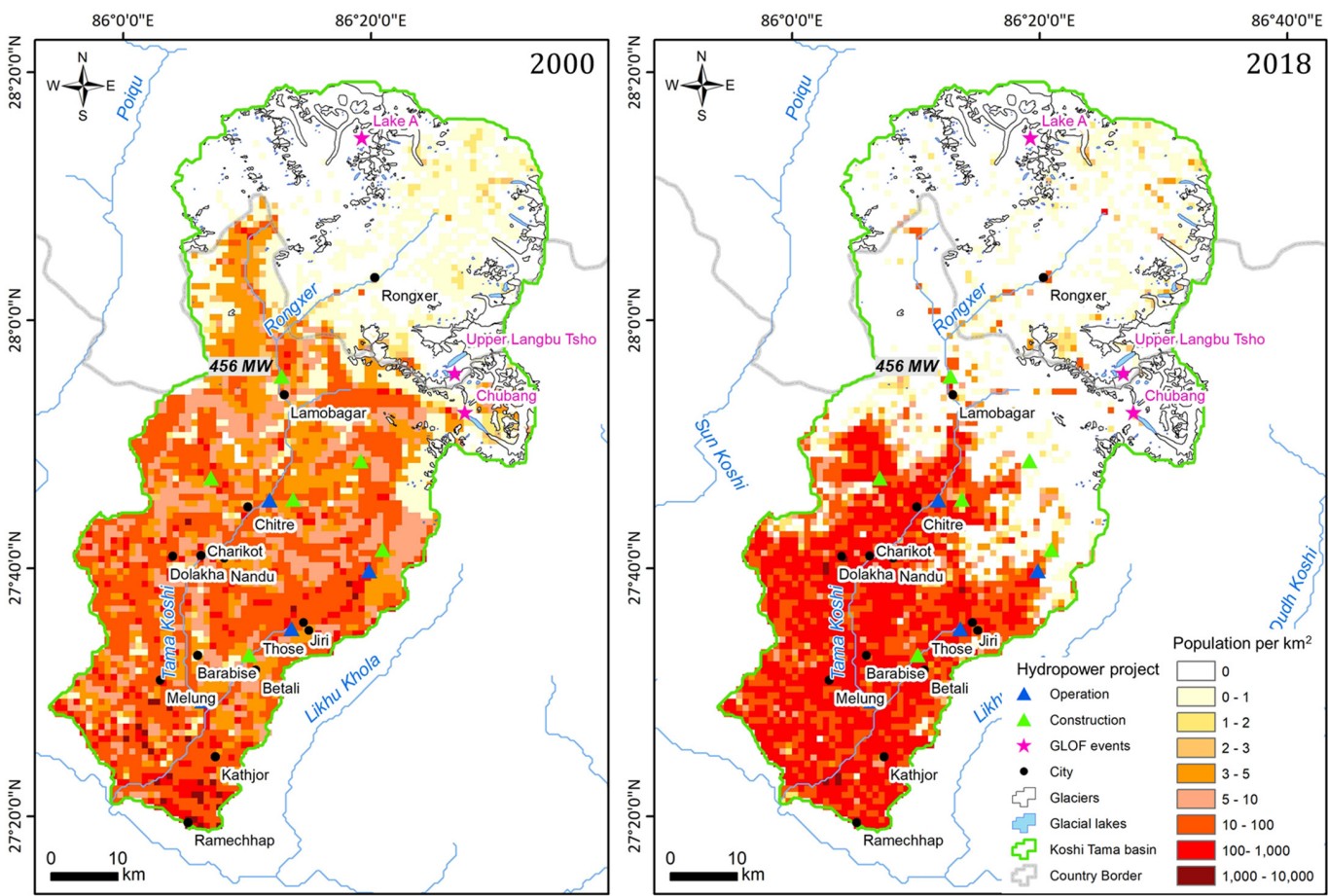

**Figure 8.** GLOF events, hydropower projects, and population tendency (left 2000 and right 2018) in the basin.

**Table 6.** Historical GLOF events of the basin.

| Name | Date of Outburst | Location | (Possible) Trigger | Source |
|---|---|---|---|---|
| Chubung | 12 July 1991 | Rolwaling Valley, Nepal | Ice avalanche | (Reynolds 1999) |
| Upper Langbu Tsho | 22 September 1992~ 17 November 1992 | Dagazhuoma River, China | Ice avalanche | (Nie et al., 2018) |
| Lake A | 30 July 2018~9 August 2018 | Labujikongzangbu River, China | Extreme precipitation | This study |

With the increase in human activities (the population grew from 34 per km² in 2000 to 59 per km² in 2018, Figure 8) and socioeconomic development in both Nepal and Chinese parts of the Tama Koshi basin, it is demanded to construct hydropower facilities to reduce the dependence on biofuels and fossil fuels imports. At present, approximately 10 hydropower projects (the largest hydropower project has an installed capacity of 456 MW, which is also the largest hydropower project of Nepal) in the lower reaches of Tama Koshi permits have been issued, of which 6 are under construction (Figure 8). However, eight rapidly expanding glacial lakes are developed in the basin, many of which are close to the roads, village, and other infrastructures, the straight-line distance between them and

the nearest hydropower station is about 15 km. Future climate warming will amplify the consequence and increase the risk of GLOFs hazard [14,40,42,70]. Meanwhile, with the improved living standard and increased public awareness of disaster prevention and mitigation (increased awareness of GLOF hazards) in the Tama Koshi basin, numerous residents have migrated from rural areas (the upper valley) to urban areas, resulting in a significant increase in population density in downstream cities (Figure 8). This avoids some small-scale GLOFs but increases the effect when large-scale GLOFs occur. For example, a recent study [60] has shown that the Tsho Rolpa lake elevation has increased in recent years. If the lake area gradually increases, as we expect (see Section 5.3), the water pressure on the dam body will be greatly increased. Once a large-scale outburst occurs, it will bring disastrous effects to the highly populated areas in Lamobagar.

Remote sensing is a vital tool in quickly monitor the glacial dynamics and hazard risk assessment [71]. While this study has obtained detailed information on glacial lakes area changes, basic data on glacial lakes (e.g., lake basin topography, type of dam construction material, the internal structure of moraine dams, etc.) are still lacking and field investigations are needed. To date, the lake basin parameters and dam characteristics were measured only at Tsho Rolpa in Nepal [39]. Detail investigations and research are relatively rarely conducted in the Chinese Tibet part, where some glacial lakes with rapid expansion and their GLOFs could lead to transnational impacts on downstream Nepal. For those lakes with high risks (e.g., Niangzongmajue and Yanongcuo in China), hazard assessment is crucial, and, if necessary, artificial drainage measurements could be adopted to reduce their GLOF risks. For example, Tsho Rolpa in Nepal has been automatically drained by the construction of an open channel and siphon since 1995, and its outburst risk has been basically controlled, and even keep stable during the Gorkha earthquake in 2015 [72].

## 6. Conclusions and Remarks

Using 42 moderate- and high-resolution remote sensing images between 1964 and 2020, we completed a comprehensive investigation of glaciers and glacial lakes of the entire Tama Koshi basin, central Himalayas. We mapped a total of 271 glaciers and 196 glacial lakes with coverage of $329.2 \pm 1.9$ km$^2$ and $14.4 \pm 0.3$ km$^2$, respectively, in 2020 across the basin. During the past 56 years, the total area of glaciers in the basin has shrunk by $26.2 \pm 3.2$ km$^2$ (0.13% a$^{-1}$) and thinned by ~20 m on average, with their mean velocity has gradually slowed down from 5.3 m a$^{-1}$ in 1999–2003 to 4.0 m a$^{-1}$ in 2013–2015. The total area of glacial lakes increased by $9.2 \pm 0.4$ km$^2$ (~180%), with ice-contacted proglacial lakes showing a much higher expansion rate (~204%) than the others. Moreover, large-scale glacial lakes are developed preferentially and experienced rapid expansion on the east side of the basin. For land-terminating debris-covered glaciers, the decrease in ice thickness is the main cause of its mass loss. However, for lake-terminating glaciers, rapid calving retreat of the terminus was the dominant process. Preliminary analysis shows that the increase in temperature and precipitation caused the glacier shrinkage and glacial lake expansion in the Tama Koshi basin. Meanwhile, topography also acted as a vital factor that not only affects the expansion of glacial lakes but also controls the development of glacial lakes. We hypothesize that lake expansion will continue in some cases until the critical local topography of steepening icefall is reached, but the number of lakes may not necessarily increase. Many residents (~58 per km$^2$ in 2018) and hydropower stations (~10 in 2020) are sited in the lower reaches of the basin. The impact of the hazard will be amplified if a transnational GLOF occurs. To better assess and reduce the GLOF risks, further detailed monitoring should be focused on the eight rapidly expanding proglacial lakes in the upper reaches in Chinese Tibet Rongxer, e.g., obtaining the basic information (such as lake bathymetry, lake area/level fluctuation, dam characteristics and downstream high-resolution terrain) of these lakes via field and in situ investigations.

**Supplementary Materials:** The following are available online at https://www.mdpi.com/article/10.3390/rs13183614/s1, Figure S1: An example of extracting the rapid expansion glacial lakes on Landsat images using Google Earth Engine Python API, Figure S2: Identified Lake A GLOF event by

PlanetScope imagery, Figure S3: Supraglacial lakes and ice cliffs on Glacier M base on PlanetScope image, Table S1: Data used for this study.

**Author Contributions:** Conceptualization, Q.L.; Data curation, Y.Z.; Formal analysis, Y.Z; Funding acquisition, Q.L. and Y.W.; Investigation, Y.Z.; Methodology, Y.Z.; Project administration, Q.L. and Y.W.; Visualization, Y.Z.; Writing—original draft, Y.Z.; Writing—review and editing, Y.Z., Q.L., L.S., Y.L., H.W., H.L. and Y.W. All authors have read and agreed to the published version of the manuscript.

**Funding:** This research was funded by the "Chinese Academy of Sciences Overseas Institutions Platform Project, grant number 131C11KYSB20200033" and the "IAEA Interregional Technical Co-operation Project, grant number INT/5/156"; Our research was also supported by the "National Natural Science Foundation of China Project, grant number 41871069" and the "Sichuan Science and Technology Programs, grant number 2021JDJQ0009 and 2020JDJQ0002".

**Data Availability Statement:** The data sets of the glacier and glacial lake in the Tama Koshi basin and the Python code for this study can be found and downloaded from https://doi.org/10.5281/zenodo.5496959.

**Acknowledgments:** The authors gratefully acknowledge the PlanetLab for the provision of PlanetScope imagery and U.S. Geological Survey for Landsat and Keyhole images, and the authors would also like to thank developers in the GEE community and Qiusheng Wu from the University of Tennessee for sharing very helpful sample codes.

**Conflicts of Interest:** The authors declare no conflict of interest.

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
