# Peer review of "Rapid glacier Shrinkage and Glacial Lake Expansion of a China-Nepal Transboundary Catchment in the Central Himalayas, between 1964 and 2020"

_remotesensing, doi:10.3390/rs13183614_

Round 1
Reviewer 1 Report
I consider the submitted article to be of very high quality. From the introduction to the issue, through the data used to the processing and discussion of the results. In one table we have marked one value, which I think is not correct. I did not find any other shortcomings in the article. I agree with the authors with their statement in the conclusion that it is necessary to perform a detailed mapping of the most dangerous lakes directly in the field. If they do this, I wish you good luck and look forward to the results of this detailed in-situ research.

Author Response
Dear Reviewer,
Thank you for taking the time to review our manuscript and affirm our research. We have corrected the number in Table 4, thanks for mention.
With kind regards,
On behalf of all authors
Yan Zhong and Qiao Liu
Reviewer 2 Report
The authors provide a detailed assessment of glacier change and glacial lake change in the Tama Koshi Basin. The scientific methods are appropriate and well described. Comparisons with previous studies need to be expanded in the discussion, this will help identify the additional value of the study. Figure 7 and Figure 9 are both useful additions, improvements in Figure 7 legend and better quantification of the Hydropower station distances from glaciers and MW potential will add further value to the paper. There is space to separate the discussion of lakes A-H and their associated glaciers into separate paragraphs.
61: “To better assess and mitigate….”
76: “where the Tama Koshi joins the Sunkoshi, to above 8000 m a.s.l. adjacent to Mt. Cho Oyu (8188 m a.s.l.).”
92: Rounce et al (2017) noted 31 high risk lakes and 11 very high risk lakes in Nepal. This should be cited along with how many were in the Tama Koshi.
144: “geometric correction of the distorted image is completed using coordinate transformation.”
173: ” Subsequently, equidistant points at a 30 m spacing were generated along the entire glacier center flow line to identify and visualize the elevation, flow velocity, and slope.”
195: “southern oriented glaciers have the maximum area coverage (84.9±0.4 km2, ~26%, Figure 2C). Northwest oriented glaciers have…”
206: This will automatically happen due to the orientation of valleys on the respectively east and west side of a basin. This is not useful information. “the glaciers in the east have a more southwest orientation while in the west they are more oriented to the southeast.”
222: The mean retreat rate can be influenced by a few glaciers, what is median for land based and lake terminating?
264: Not clear what the difference is between the 196 glacial lakes and 107 glacial lakes?
289: “the total number of moraine-dammed ice-contacted lakes peaked around 1980 at 17 an increase from 10 in 1964. After 1980, the number of moraine-dammed ice-contacted glacial lakes gradually decreased to 5 in 2020, the other 12 were disconnected from the retreating mother glacier and will no longer expand.”
300: Given that small lakes under 0.05 km2 have limited volume and hence limited potential impact. The smallest lakes are also more transient and more impacted by image resolution. Given that greater focus should be on the larger glacial lakes.
308: Just because they are rapidly expanding why identify them as PDGLs?
310: This references the Lake G glacier basin? In this section it is better to refer to the lakes/glacier by its actual name where it exists vs a letter.
314: Did Lake A burst or simply drain?
323: How does your assessment of Tsho Rolpa compare to the many other assessments?
337: It is crucial given this basin has been examined before in terms of glacial lakes to compare your results to a couple of pertinent studies. Khadka et al (2018) looks at a number of the same parameters as utilized here, though focused not just on the Tama Koshi. How do your observations compare?
349: The freezing level rise is of comparable importance to the temperature rise. Note Pelto et al (2021) document this for Everest area during the 1950-2021 period along with the resultant rising snow lines.
357: Xiang et al (2014) point out the much greater debris cover in Tibet than Nepal in the basin. The debris covered area in Nepal is limited with 4.3 km2 or 8% and the debris covered area of glaciers in the Poiqu Basin in China is assessed at 51 km2 of debris covered ice.
364: Your data suggests like other studies that lake terminating glaciers are losing mass more rapidly than land terminating. However, this does not account for most of the loss, since the difference between the two in your and other studies is less than 50% of the total loss.
386: How does this slope compare to slope of non-lake terminating glaciers?
Figure 7: Need to better identify which lines are elevation and which are velocity in legend.
438: “lower reaches of Tama Koshi permits have been issued”
441: You have indicated there has been one GLOF since 1992 in the basin, in 2018 that did not cause any noted damage, how does this support increased risk of GLOF?
484: “Many residents (~58 per km2 in 2018) and hydropower stations (~10 in 2020) are sited in the lower reaches of the basin.” How far from the glacial lakes are the nearest stations?
Khadka, N.; Zhang, G.; Thakuri, S. Glacial Lakes in the Nepal Himalaya: Inventory and Decadal Dynamics (1977–2017). Remote Sens. 2018, 10, 1913. https://doi.org/10.3390/rs10121913
Pelto, M.; Panday, P.; Matthews, T.; Maurer, J.; Perry, L.B. Observations of Winter Ablation on Glaciers in the Mount Everest Region in 2020–2021. Remote Sens. 2021, 13, 2692. https://doi.org/10.3390/rs13142692
Rounce, D.R.; Watson, C.S.; McKinney, D.C. Identification of Hazard and Risk for Glacial Lakes in the Nepal Himalaya Using Satellite Imagery from 2000–2015. Remote Sens. 2017, 9, 654. https://doi.org/10.3390/rs9070654
Xiang, Y., Gao, Y. and Yao, T. Glacier change in the Poiqu River basin inferred from Landsat data from 1975 to 2010. Quaternary International 2014, 349, 392–401.
Author Response
Please see the PDF file.

Reviewer 3 Report
The authors present a study which assesses the extent of glaciers and glacial lakes in a catchment in the Central Himalaya between 1964 and 2020.
The study is interesting to read, and albeit the methodology used is quite standard and not novel, the findings are worth publishing. However, I do have some remarks, which are detailed below.
General comments:
- I find the results section somewhat difficult to follow, mostly because some figures take quite some time to understand (e.g. Fig. 2). While the structure of the section (useful headings) helps, the paragraphs are a little bit confusing at times, and the core results / messages are somewhat blurred. At the end of section 4, I found myself asking what the core message is that the authors wanted to convey. A large focus is pure on exploratory results, and I think these should be better supported in terms of data visualization. Maybe the authors could slightly streamline the section with a focus on the main take-home-messages that are later discussed in (a well written) section 5.
- I strongly advise to consistent colors in figures throughout the manuscript. See https://colorbrewer2.org/ or http://colorspace.r-forge.r-project.org/ for details on reader-friendly color palettes. All plots look slightly different in terms of choice of colors. Personally, I prefer a consistent design pattern and look in all plots throughout a manuscript.
Specific comments:
- Line 61: The authors could consider referencing Avian et al. (2020) provide a recent overview of earth observation techniques for monitoring high mountain environments (https://doi.org/10.3390/rs12081251).
- Line 80: Please specify the source of the temperature/precipitation data. Are these station measurements, reanalysis data, or some other sort of gridded/interpolated station data? What is the temporal basis for the means? (e.g. 1990-2020)? This is mentioned later in 3.4, but please clarify here as well.
- Line 97: inconsistent citation (Sakai., 2020)
- Figure 1: (A) Please add an explanation of the letters to the caption, this follows only latter in the text. "Letters indicate locations of glacier lakes" or something will suffice. (B) The use of colors is counter-intuitive to me. Suggest to use blue for winter, red for summer, and black for mean. Also, the width of the inlay is quite small, suggest to increase the width or use boxplots for groups of years instead (otherwise we would lose the information of the increasing temperature trend, which seems to be present in the data).
- Line 110ff: Just out of curiosity: What about Sentinel-2 data? (e.g. Williamson et al. 2018; https://tc.copernicus.org/articles/12/3045/2018/).
- Line 110ff: Again, just out of curiosity: What about SAR-data?
- Line 117: What about seasonal changes in lake size (e.g. spring versus fall)? Do these play a role here?
- Line 121ff: This paragraph is somewhat difficult to read and follow. Maybe use a table of data sets?
- Line 153: What does "complemented with manual modifications" mean? Please clarify.
- Line 187: Remove "that", or rephrase sentence.
- Figure 2: (A) I strongly suggest to use a simple transfer function when using dual y axis (e.g. a factor of 30, as opposed to 450/16 = 28) otherwise plots are difficult to read. The y-axis ticks of the right y-axis do not correspond to the grid lines of the left y-axis. (B) What is the unit of the top x-axis?
- Figure 3: Please add a title to the legends (change rates and velocities), is not immediately clear what is plotted. Please add units (%/a and m/a) as well.
- Table 3: "Number" is a somewhat ambiguous column name, please clarify.
- Line 259: 4.1 should read 4.2
- Figure 4: Suggest to use a consistent appearance, with figure labels at the same position. The colors are extreme difficult to differentiate (e.g. 2005, 2006, 2016, 2017), suggest to use continuous color palette here.
- Figure 5: Again, I suggest to stick to a consistent color scheme. Please increase font size of the legend to improve readability.
- Line 339: Since humans have an impact on climate change, one could argue that glacial lakes are influenced by humans. I suggest to replace "humanity" with "direct human activities" or similar.
- Line 384: please add this information to the caption of Fig. 7.
- Fig. 7: again, dual y-axis, different panel size of the plots makes this somewhat difficult to read. The meaning of the pink area is not explained in the caption. What is the main difference that we can see between lake-terminating and land-terminating glaciers here?
- Figure 8: Do the colors have any meaning? If not, I suggest to simply use one color. Are the plotted points the underlying samples of the boxplots? Using the most common convention, whiskers should end at observation points, with a maximum length of 1.5 IQR. everything else is plotted as outliers. However, most whiskers do not end at data points. Finally, what do the crosses mean? Could be the mean value, but this is also not clear.
Author Response
Please see the PDF file.

Reviewer 4 Report
The authors did a lot of work to reveal the tendencies of development of the glacial lakes and the changes of the glaciers in a highly glacierized mountainous area with a large risk of GLOFs, affecting the populated areas, which means not only scientific but the potential practical value of the results of their work. The obtained results and conclusions are convincing, but I would like to more clearly establish the relationship between climatic changes and the changes in the rates of degradation of glaciers and an increase in the areas of lakes recorded by the authors. There are also small comments on pictures and text.
- Table 1 shows a decrease in the rate of reduction of glaciers in 1990-2010, correlating with a slowdown in lake areas increase (table 3)? What is the reason for this? It seems, that there is no correlation with the rates of changes in temperature (table 3).
- Figure 3 - it is necessary to indicate the units of measurement and show which figures relate to the elevation of the glaciers, and which to their velocities
- Figure 8 is unclear- what do the points and the colored rectangles mean? Please, indicate it in the legend
- Line 422 ..and caused the lake to completely drained… - correct the grammar
Author Response
Please see the PDF file.
